# Escaping Policy Contraction: Contraction-Aware PPO (CaPPO) for Stable Language Model Fine-Tuning

**Dun Yuan**
School of Computer Science
McGill University
Montreal, QC, Canada
`dun.yuan@mail.mcgill.ca`

**Di Wu**
Department of Mechanical, Industrial
and Aerospace Engineering
Concordia University
Montreal, QC, Canada
`di.wu@concordia.ca`

**Xue Liu**
Mohamed bin Zayed University of Artificial Intelligence
Abu Dhabi, UAE
School of Computer Science
McGill University
Montreal, QC, Canada
`xue.liu@mbzuai.ac.ae`

## Abstract

Reinforcement learning from human feedback (RLHF) with proximal policy optimization (PPO) is widely used but often yields less diverse outputs than supervised fine-tuning, suggesting an effect in which the policy's support contracts during on-policy optimization. We formalize this "policy contraction" with the Support Retention Ratio (SRR)—the share of SFT completions that retain non-negligible probability under the RL policy—and additionally track token-entropy, Kullback–Leibler (KL) divergence to the reference, and repetition. We propose Contraction-Aware PPO (CaPPO), a minimum-norm multi-gradient update that co-optimizes reward, entropy, and KL, paired with a controller that steers exploration toward a target token entropy. On HH-RLHF, Summarize-from-Feedback, and UltraFeedback with Qwen2-7B, Qwen2.5-14B, Mistral-7B-Instruct, and Llama-3-8B-Instruct, CaPPO increases win rate by 2 to 4 points over PPO and improves diversity, gaining 0.2 to 0.3 higher SRR. The gains persist under decoding sweeps and are robust to reward scaling and critic variance. Treating reward, diversity, and stability as first-class objectives, CaPPO mitigates contraction without sacrificing alignment performance.

## 1 Introduction

Reinforcement learning from human feedback (RLHF) (Christiano et al., 2017) has become a standard paradigm for adapting supervised language models to human preferences. A typical pipeline trains a preference-based reward model and then optimizes the policy with Proximal Policy Optimization (PPO) (Schulman et al., 2017) while regularizing toward a supervised reference via a Kullback–Leibler (KL) penalty to stabilize training and curb reward hacking (Stiennon et al., 2020; Ouyang et al., 2022). This recipe underpins influential alignment systems in instruction following and dialogue, and annotation costs have been reduced through AI-assisted variants (Glaese et al., 2022; Bai et al., 2022b; Lee et al., 2023). Recent studies further suggest RL-based post-training can enhance language understanding and strengthen reasoning-centric models (Hu et al., 2024; Guo et al., 2025). Despite these successes, practitioners frequently observe a decline in output diversity during on-policy fine-tuning: per-token entropy falls, repetitions increase, and many plausible alternatives under the reference model receive negligible probability.

We term this phenomenon **policy contraction**: the policy's support progressively narrows as probability mass concentrates on a shrinking subset of completions—even as reward or win rate improves. Conventional diversity proxies, including Self-BLEU (Zhu et al., 2018) and Distinct-$n$, offer useful signals but are decoding-sensitive and can be noisy (Holtzman et al., 2019; Li et al., 2015; Celikyilmaz et al., 2020). To probe support loss more directly, we use the Support Retention Ratio (SRR): the fraction of supervised completions that retain non-negligible length-normalized log-likelihood under the trained policy at a fixed threshold. SRR measures how much of the supervised set remains likely under the trained policy. It counts the fraction of supervised completions whose length-normalized log-likelihood exceeds a percentile threshold set from the reference. This metric is independent of decoding and comparable across prompts.

This perspective clarifies how different post-training choices shape distributional behavior. RL-free preference-optimization methods such as Direct Preference Optimization (DPO) and Kahneman-Tversky Optimization (KTO) streamline optimization and often preserve more of the SFT distribution, but they operate off-policy and do not directly address on-policy distributional drift that emerges in RLHF (Rafailov et al., 2023; Ethayarajh et al., 2024). PPO variants that target better credit assignment or critic robustness, like VinePPO and group-normalized approaches used for reasoning, raise performance but may still emphasize a small set of high-reward modes without explicit diversity control (Kazemnejad et al., 2024; Mroueh, 2025). Recent analyses in the verifiable-reward setting suggest that reinforcement learning can predominantly reweight solution-bearing modes rather than expand them, which aligns with the contraction picture (Yue et al., 2025). At the same time, more general evidence indicates that RL can improve language understanding in NLU tasks (Hu et al., 2024) and enable strong reasoning systems (Guo et al., 2025), underscoring that addressing contraction is a distributional problem, not a purely capability problem.

We address contraction by elevating diversity control from an auxiliary penalty to a core training objective. Standard PPO implicitly prioritizes reward maximization, using entropy and KL regularization as secondary terms with fixed or manually tuned weights. This scalarization makes the optimization brittle: when the reward scale is large, or critic estimates are noisy, entropy rapidly collapses and the policy contracts onto a narrow set of completions. Once entropy falls below a certain level, exploration becomes ineffective, repetitions rise, and the probability mass shifts disproportionately to a small subset of high-reward outputs. These dynamics explain why PPO-trained models often exhibit rising repeat rates, declining support retention, and left-shifted log-probability histograms, even when alignment reward improves. Controlling this collapse with static entropy coefficients or manually adjusted KL weights is unreliable and sensitive to dataset, backbone, and scale.

To overcome these limitations, we propose **Contraction-Aware PPO (CaPPO)**. CaPPO reinterprets reward, entropy, and KL-to-reference as peer objectives and computes parameter updates with a minimum-norm multi-gradient descent procedure that approximates a Pareto-improving step (Parisi et al., 2014; Van Moffaert et al., 2013; Momma et al., 2022; Liu & Vicente, 2024). This avoids brittle scalarization and ensures that progress on reward does not come at the expense of collapsing entropy or uncontrolled KL drift. In addition, CaPPO introduces an entropy-scheduling controller that monitors per-token entropy and dynamically adjusts the effective entropy coefficient: injecting exploration pressure when entropy collapses and relaxing it when entropy is sufficient. Together, these two components yield a method fully compatible with PPO's clipped surrogate, which directly mitigates contraction. Empirically, CaPPO restores support retention and improves diversity metrics while matching or exceeding task reward and win rate across different models and datasets. Its improvements are consistent across seeds, reward scales, and critic variances, highlighting robustness and confirming that treating entropy and KL as first-class objectives is essential for preserving diversity and accuracy in RLHF.

The main contributions of this work are as follows:

- We verify the existence of policy contraction in PPO-based RLHF and introduce SRR as a direct, decoding-agnostic measure of support preservation.
- We present CaPPO with entropy scheduling as a drop-in extension that balances reward maximization with KL control and entropy maintenance through a Pareto-motivated multi-objective update.

- We empirically verify contraction and show that CaPPO mitigates it across Qwen, Llama, and Mistral backbones on HH-RLHF, Summarize-from-Feedback, and UltraFeedback, improving SRR and diversity at competitive or higher win rates. We also discuss connections to overtraining and post-training adaptability.

The remainder of the paper proceeds as follows. Section 2 reviews RLHF foundations, RL-free preference optimization, degeneration/diversity metrics, and multi-objective optimization. Section 3 quantifies contraction using sampling-based diversity, entropy/KL trajectories, and SRR with log-probability histograms of SFT completions. Section 4 details CaPPO and the entropy-scheduling controller, including optimization geometry and practical solvers. Section 5 reports main results, ablations, robustness to reward scaling and critic variance, and decoding-independence analyses, before Section 6 outlines limitations and future directions.

## 2 RELATED WORK

**RLHF and PPO.** Modern alignment pipelines pair preference-trained rewards with PPO under KL regularization to a reference model (Schulman et al., 2017; Christiano et al., 2017; Stiennon et al., 2020; Ouyang et al., 2022; Glaese et al., 2022). This recipe improves instruction following and dialogue while mitigating reward hacking via KL-to-reference anchoring. Constitutional AI and RLAIF reduce human labelling by distilling guardrails and using AI feedback, respectively, yet retain a preference/RL phase (Bai et al., 2022b; Lee et al., 2023). Control-theoretic views of KL-regularized RL explain how anchoring shapes exploration and stability under information constraints (Galashov et al., 2019). Recent evidence shows RL-based post-training can also strengthen NLU (Hu et al., 2024) and enable state-of-the-art reasoning systems (Guo et al., 2025). Our work focuses on a distributional side effect of this pipeline—policy contraction—by measuring support loss directly and proposing a contraction-aware modification of PPO that optimizes reward, entropy, and KL as peer objectives.

**Reasoning with RL.** Reasoning workloads surface weaknesses in token/value credit assignment and critic stability. VinePPO addresses step-wise credit propagation for multi-step reasoning, improving RL training quality under PPO (Kazemnejad et al., 2024). In parallel, GRPO-style critic-free training with group normalization has seen broad adoption in reasoning systems; its effective loss and dynamics can be written as a KL-regularized contrastive objective with success amplification guarantees (Mroueh, 2025), and large-scale instances demonstrate strong math/code gains (Guo et al., 2025). However, recent analyses indicate that such RL procedures often reweight solution-bearing modes rather than expand the underlying support, improving pass@1 while narrowing the reachable set at larger sample budgets (Yue et al., 2025). This view is consistent with our findings: standard PPO can raise reward or win rate even as entropy declines, repetition rises, and the log-probability mass concentrates on fewer completions.

**RL-free alignment.** RL-free preference optimization aims to stabilize and simplify post-training by sidestepping on-policy updates. DPO matches the optimal RLHF policy under a specific reward parameterization via a simple preference-classification loss (Rafailov et al., 2023); KTO reframes alignment as prospect-theoretic optimization to better capture asymmetric human preferences (Ethayarajh et al., 2024); and active preference optimization improves sample efficiency by selecting informative prompts (Das et al., 2024). These methods often preserve more of the SFT distribution and reduce engineering overhead. However, they operate off-policy and do not directly control the on-policy distributional drift that emerges during RLHF training.

**Diversity and multi-objective methods.** Text generation is prone to degeneration and mode collapse under likelihood-centric training and decoding (Holtzman et al., 2019), motivating diversity-oriented objectives (e.g., promoting dissimilar responses) and evaluation metrics such as Self-BLEU and Distinct-$n$ (Li et al., 2015; Zhu et al., 2018; Celikyilmaz et al., 2020; Montahaei et al., 2019). Recent analyses in SFT emphasize preserving support via reverse-KL with entropy, which can improve generalization and test-time scaling (Li et al., 2024). Balancing reward maximization with KL anchoring and entropy maintenance is inherently multi-objective; classical MORL offers policy-gradient methods and scalarization critiques (Parisi et al., 2014; Van Moffaert et al., 2013), while

Pareto-stationary multi-gradient updates provide principled directions in ML and RL (Momma et al., 2022; Liu & Vicente, 2024; Lin et al., 2022; Liu et al., 2025; Kang et al., 2024). Related contemporaneous observations on overtraining suggest that heavily pre-trained models can be more complex to adapt (Springer et al., 2025), reinforcing the need for post-training procedures that preserve support rather than further narrow it.

## 3 POLICY CONTRACTION

**Background and motivation.** Preference-based fine-tuning with PPO has repeatedly been observed to narrow a model's output distribution: as optimization proceeds, probability mass concentrates on a few high-reward modes, token-level entropy falls, and many plausible alternatives under the SFT/reference policy receive vanishing probability. Prior analyses discuss reward over-optimization, KL regularization effects, and decoding sensitivity, but a clean, decoding-agnostic diagnosis of support loss has been lacking. This section establishes that contraction appears across models and prompts under controlled conditions.

**Setup and principles.** We aim to quantify support loss independently of decoding heuristics and reward-scale artifacts. We therefore report multiple random seeds, sweep standard decoding settings in a separate robustness check, and fix reward scaling where noted. We probe contraction using three complementary diagnostics:

**(V1) Sampling-based diversity.** We sample $k$ completions for each prompt and compute Self-BLEU, Distinct-$n$, and $n$-gram repetition rate.

**(V2) Entropy and KL trajectories.** We track length-normalized per-token entropy and forward KL to the reference over PPO iterations:

$$H = \mathbb{E}_{x,\, y\sim\pi_\theta(\cdot|x)}\Big[ -\tfrac{1}{|y|}\log\pi_\theta(y\mid x)\Big], \qquad \mathrm{KL}(\pi_\theta \,\|\, \pi_{\mathrm{ref}}).$$

Contraction is indicated by sustained entropy decline at fixed or rising KL.

**(V3) Support Retention Ratio (SRR).** We estimate

$$\mathrm{SRR}(\tau) = \mathbb{E}_x \Pr_{y\sim\pi_{\mathrm{ref}}(\cdot|x)}\Big[\tfrac{1}{|y|}\log\pi_\theta(y\mid x) \geq \tau\Big],$$

the fraction of SFT completions whose length-normalized log-probability under the PPO policy exceeds a fixed threshold $\tau$.

Table 1: Distributional diagnostics on verification sets. Under PPO, entropy drops while forward KL to the reference rises—a contraction signature. Entropy scheduling recovers entropy and reduces repetition with only modest KL change. Repetition is reported as % of repeated bigrams; entropy is in nats/token; KL is in nats.

| Model | Dataset | Setup | Repeat-2 | Entropy | KL to ref |
|---|---|---|---|---|---|
| Llama3-8B | ShareGPT-1K | SFT-only | 21.8 | 3.88 | — |
| | | + PPO | 24.5 | 3.42 | 0.45 |
| | | + Entropy | 18.0 | 3.76 | 0.43 |
| Qwen2-7B | AlpacaEval | SFT-only | 21.1 | 3.90 | — |
| | | + PPO | 22.7 | 3.50 | 0.46 |
| | | + Entropy | 18.7 | 3.77 | 0.44 |
| Qwen2-7B | ShareGPT-1K | SFT-only | 20.0 | 3.84 | — |
| | | + PPO | 23.1 | 3.33 | 0.45 |
| | | + Entropy | 17.1 | 3.70 | 0.43 |

We run (V1)–(V2) on ShareGPT-1K and AlpacaEval; for external validity with preference data, (V3) is reported on ShareGPT-1K, AlpacaEval, and HH-RLHF. Table 1 shows that PPO reduces entropy while KL rises, for example $3.88 \rightarrow 3.42$ and $3.90 \rightarrow 3.50$ nats/token as KL increases to $\approx 0.45$–$0.46$ nats, consistent with contraction. Entropy scheduling reverses the entropy dip and lowers repetition with only modest KL change.

Table 2: Test of policy contraction. Fraction of SFT completions retained above threshold $\tau$. SRR increases substantially with entropy scheduling.

| Dataset | Setup | SRR@$\tau$ |
|---|---|---|
| ShareGPT-1K | + PPO | 0.37 |
| AlpacaEval | + PPO | 0.39 |
| HH-RLHF | + PPO | 0.41 |
| ShareGPT-1K | + Entropy | 0.58 |
| AlpacaEval | + Entropy | 0.60 |
| HH-RLHF | + Entropy | 0.62 |

**Qualitative density shift.** Log-probability histograms over SFT-sampled completions show post-PPO mass concentrating on fewer sequences, with a heavier left tail (near-zero probabilities) on many previously feasible modes. Entropy scheduling broadens the histogram and increases SRR, indicating improved support preservation.

Across ShareGPT-1K and AlpacaEval, we observe consistent entropy declines and low SRR under PPO. Diversity-aware control (§4) restores entropy, reduces repetition, and increases SRR, motivating the multi-objective design, which will be elaborated next.

## 4 METHODOLOGY

We aim to mitigate policy contraction in PPO-based RLHF, where the policy's support narrows over training, reducing entropy and diversity. We introduce two complementary components: (i) CaPPO, which treats reward, entropy, and KL as peer objectives and selects a Pareto-improving update at each step; and (ii) an entropy-scheduling controller that stabilizes exploration by steering token entropy toward a target. Throughout, $\pi_{\text{ref}}$ denotes the SFT reference policy, $\pi_\theta$ the trainable policy, and $\mathcal{R}(x, y)$ a preference-model score. Sequence log-likelihoods are length-normalized unless otherwise stated.

### 4.1 RLHF SETUP AND PPO SURROGATE

Given prompts $x$, we sample completions $y$ from $\pi_\theta$, score them with $\mathcal{R}$, and diffuse sequence reward across tokens for advantage estimation: $r_t = \mathcal{R}(x, y)/|y|$, with advantages $A_t$ via GAE. The clipped PPO surrogate on the reward component is

$$\mathcal{L}_{\text{reward}}^{\text{PPO}}(\theta) = \mathbb{E}_t\Big[ \min\big(\rho_t(\theta)A_t,\ \text{clip}(\rho_t(\theta), 1 - \epsilon, 1 + \epsilon)A_t\big)\Big], \quad \rho_t(\theta) = \frac{\pi_\theta(a_t \mid s_t)}{\pi_{\theta_{\text{old}}}(a_t \mid s_t)}. \quad (1)$$

### 4.2 CONTRACTION-AWARE PPO (CAPPO)

Standard practice augments equation 1 with fixed coefficients on entropy and KL. Such scalarization is brittle: overweighting reward accelerates contraction by eroding entropy and drifting from $\pi_{\text{ref}}$, while overweighting regularizers suppresses reward learning. CaPPO instead searches at every step for a descent direction that simultaneously respects reward improvement and support preservation by solving a small multi-objective problem.

We define three maximization objectives

$$J_{\text{r}}(\theta) = -\mathcal{L}_{\text{reward}}^{\text{PPO}}(\theta), \qquad J_{\text{e}}(\theta) = H(\pi_\theta), \qquad J_{\text{kl}}(\theta) = -\text{KL}\big(\pi_\theta(\cdot \mid x) \,\|\, \pi_{\text{ref}}(\cdot \mid x)\big),$$

with gradients $g_{\text{r}} = \nabla J_{\text{r}}$, $g_{\text{e}} = \nabla J_{\text{e}}$, and $g_{\text{kl}} = \nabla J_{\text{kl}}$. A point $\theta^\star$ is Pareto stationary if the origin lies in the convex hull of these gradients:

$$\mathbf{0} \in \text{co}\{\, \nabla J_{\text{r}}(\theta^\star),\ \nabla J_{\text{e}}(\theta^\star),\ \nabla J_{\text{kl}}(\theta^\star) \,\}. \quad (2)$$

We approximate equation 2 by finding the minimum-norm convex combination of gradients:

$$\min_{\lambda \in \Delta_3} \|\lambda_{\text{r}} g_{\text{r}} + \lambda_{\text{e}} g_{\text{e}} + \lambda_{\text{kl}} g_{\text{kl}}\|_2^2, \qquad \hat{g} = \sum_i \lambda_i g_i, \quad \theta \leftarrow \theta + \eta_\theta \hat{g}. \quad (3)$$

Because objectives differ in scale, we precondition with a diagonal metric $P^{-1/2}$ (e.g., Adam's second moment on $g_r$ or a Fisher diagonal from $\pi_{\text{ref}}$), solve equation 3 in the preconditioned space, and map back. A simpler alternative is unit-length normalization of $g_i$. For three objectives, the problem is reduced to a small quadratic program; we use two or three projected-gradient or Frank–Wolfe steps and check edge solutions. A constrained view makes the connection clearer:

$$\max_{\theta} \ J_r(\theta) \quad \text{s.t.} \quad \text{KL}(\pi_\theta \| \pi_{\text{ref}}) \le \epsilon_{\text{kl}}, \ \ H(\pi_\theta) \ge \epsilon_{\text{e}}, \tag{4}$$

where the Lagrange multipliers correspond to adaptive mixing weights. A primal–dual update naturally enforces these constraints, while a guarded line search ensures reward progress without excessive entropy loss or KL growth. When gradients conflict, lower bounds on mixing weights are raised according to their cosine similarity, discouraging collapse.

---

**Algorithm 1** CaPPO with Pareto mixing (RLHF)

---

1: Initialize $\theta$, optimizer, $\lambda \leftarrow (1, 0, 0)$, $\beta \leftarrow \beta_0$, $\hat{H} \leftarrow H_\star$
2: **for** each iteration **do**
3:     Collect on-policy rollouts with $\pi_\theta$; compute advantages and rewards from $\mathcal{R}$; estimate token entropy $H_t$ and $D_{\text{KL}}(\pi_\theta \| \pi_{\text{ref}})$
4:     $\hat{H} \leftarrow (1-\rho)\hat{H} + \rho \cdot \text{mean}(H_t)$; $\beta \leftarrow \text{clip}(\beta \exp(\eta_\beta(H_\star - \hat{H})), \beta_{\min}, \beta_{\max})$
5:     $g_r \leftarrow \nabla_\theta L_{\text{PPO}}(\theta)$; $g_e \leftarrow \beta \nabla_\theta H(\pi_\theta)$; $g_{\text{kl}} \leftarrow -\nabla_\theta D_{\text{KL}}(\pi_\theta \| \pi_{\text{ref}})$
6:     $\tilde{g}_i \leftarrow P^{-1/2} g_i$ for $i \in \{r, e, kl\}$; $G_{ij} \leftarrow \tilde{g}_i^\top \tilde{g}_j$
7:     $\lambda^\star \leftarrow \arg\min_{\lambda \in \Delta_3} \lambda^\top G \lambda$
8:     $\hat{g} \leftarrow \sum_i \lambda_i^\star g_i$
9:     Find largest $\eta$ such that $D_{\text{KL}}(\pi_{\theta+\eta\hat{g}} \| \pi_{\text{ref}}) \le \varepsilon_{\text{KL}}$ and $H(\pi_{\theta+\eta\hat{g}}) \ge H_{\min}$
10:    $\theta \leftarrow \theta + \eta \hat{g}$
11: **end for**

---

### 4.3 Entropy scheduling

Entropy scheduling complements CaPPO by stabilizing exploration through a simple feedback controller on the entropy weight. We track the length-normalized sequence entropy

$$H_t \ = \ \mathbb{E}_{x, \, y \sim \pi_\theta(\cdot|x)}\left[-\frac{1}{|y|} \log \pi_\theta(y \mid x)\right], \qquad \tilde{H}_t \ = \ (1-\alpha)\,\tilde{H}_{t-1} \ + \ \alpha\, H_t, \tag{5}$$

where $\tilde{H}_t$ is an Exponential Moving Average (EMA) with smoothing $\alpha \in (0, 1]$.

We adapt the entropy coefficient $\beta$ toward a time-varying target $H_{\text{target}}(t)$ via a clipped proportional update:

$$\beta_{t+1} \ = \ \text{clip}\Big(\beta_{\min}, \ \beta_t \ + \ \eta \left[H_{\text{target}}(t) \ - \ \tilde{H}_t\right], \ \beta_{\max}\Big), \tag{6}$$

where $\eta$ is a stepsize and $(\beta_{\min}, \beta_{\max})$ bound the entropy weight. In practice, $H_{\text{target}}(t)$ can be chosen as a fixed constant, a scheduled decay (to shift from exploration to exploitation gradually), or an adaptive value defined as an EMA of the reference policy entropy plus a small offset to preserve support. We compute $H_t$ on the same minibatches used for PPO updates, averaging over prompts and sampled completions with lengths $|y|$ excluding special tokens. To dampen noise, $\tilde{H}_t$ is maintained as a bias-corrected EMA, and clipping ensures that the entropy term stays within a stable range (roughly 5–20% of the total surrogate magnitude at initialization). Optionally, a small integral term can be added to the update to further reduce oscillations, though we disable it by default.

### 4.4 Theoretical perspective

CaPPO can be analyzed as a stochastic approximation scheme for multi-objective policy optimization. Its update corresponds to the minimum-norm element of the preconditioned gradients' convex hull, which ensures convergence to Pareto-stationary points under standard Lipschitz continuity. In particular, if $\hat{g}$ is the CaPPO update, then for any feasible descent direction $d$, we have

$$\langle \hat{g}, d \rangle \ \ge \ 0 \quad \forall d \in \text{cone}\{-g_r, -g_e, -g_{\text{kl}}\}, \tag{7}$$

which formalizes that no descent direction can improve one objective without violating another.

A complementary trust-region perspective interprets CaPPO as a constrained optimizer with entropy floors and KL caps. The associated Lagrangian dynamics adaptively tune multipliers $\lambda_{\mathrm{kl}}, \lambda_{\mathrm{e}}$, yielding an update of the form

$$\theta_{t+1} \;=\; \theta_t + \eta_\theta \Big( \nabla J_{\mathrm{r}}(\theta_t) - \lambda_{\mathrm{kl},t} \nabla \mathrm{KL}(\theta_t) + \lambda_{\mathrm{e},t} \nabla H(\theta_t) \Big), \tag{8}$$

with projected dual steps ensuring feasibility. Finally, entropy scheduling can be understood as a proportional–integral controller that regulates the error $H_{\mathrm{target}} - H_t$, ensuring bounded entropy tracking while remaining compatible with the fast multi-gradient updates.

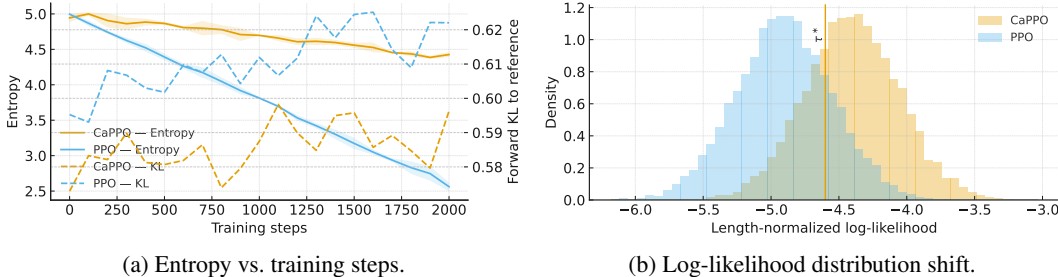

(a) Entropy vs. training steps.  (b) Log-likelihood distribution shift.

Figure 1: Policy contraction and mitigation. PPO training reduces entropy and erodes support for reference completions, while CaPPO stabilizes entropy and preserves a larger fraction of the reference distribution.

These perspectives establish CaPPO as a principled trust-region variant of PPO that preserves support while continuing to maximize reward. Empirically, this theoretical picture is borne out in Figure 1: 1a shows that PPO suffers entropy collapse at similar KL to the reference, whereas CaPPO stabilizes entropy; 1b illustrates the consequence at the sample level, where CaPPO shifts probability mass rightward and retains reference completions that PPO discards.

## 5 RESULTS

We evaluate PPO and our proposed CaPPO under a standard RLHF setup on three widely used datasets: (i) Anthropic HH-RLHF (helpfulness/harmlessness preferences) (Bai et al., 2022a), (ii) OpenAI Summarize-from-Feedback (Stiennon et al., 2020), and (iii) UltraFeedback for general helpfulness preferences (Cui et al., 2023). We compare four base models: Qwen2-7B (Team, 2024), Qwen2.5-14B (Yang et al., 2025), Mistral-7B-Instruct (Jiang et al., 2023), and Llama-3-8B-Instruct (Grattafiori et al., 2024).

**Overall performance.** Table 3 reports preference win rates relative to SFT across the three RLHF datasets and four model backbones. Several consistent observations emerge. PPO provides a clear improvement over SFT in all cases, raising the win rate by 7–15 points depending on model size and dataset. For example, Qwen2.5-14B on HH-RLHF climbs from 50.0 to $65.1\pm0.7$, while Llama-3-8B on UltraFeedback increases from 50.0 to $62.4\pm0.8$. These gains confirm the effectiveness of on-policy RL in extracting the reward-model signal. However, the pattern also shows that PPO improvements plateau at a relatively narrow margin, suggesting that further win-rate gains are difficult without additional intervention. CaPPO consistently lifts performance beyond this plateau, delivering 2–4 points higher win rate than PPO across all backbones and datasets. Importantly, these gains are not confined to a single model or dataset but appear consistently across Qwen, Llama, and Mistral families, indicating that the contraction-aware updates generalize broadly rather than exploiting a particular training recipe.

Table 4 then situates CaPPO among alternative optimization strategies. Off-policy methods such as DPO, IPO, ORPO, KTO, and RRHF preserve high SRR (up to 0.92) but underperform in win rate (62–65%). On-policy baselines like PPO, VinePPO, and GRPO push win rate higher (67–71%) but at the cost of contraction, reducing SRR to the 0.46–0.62 range. CaPPO stands out by combining the

strengths of both: it reaches the highest or co-highest win rate, lowers redundancy, increases lexical diversity, and recovers support retention.

Table 5 also provides macro-averaged evidence of how these methods shape distributional support. PPO-trained models show the characteristic contraction signature—elevated Self-BLEU (0.47–0.49), suppressed Distinct-2 (0.16–0.18), and SRR collapse below 0.45. CaPPO reverses these effects across all four backbones, lowering Self-BLEU by 0.13–0.15, raising Distinct-2 by 0.08–0.09, and improving SRR by 0.29–0.34. These gains establish that CaPPO not only achieves higher alignment reward but also broadens the effective support of the trained model, counteracting the narrowing induced by PPO alone.

Table 3: Preference win rate compared to SFT on three RLHF datasets across four base models. Means $\pm$ std over three seeds. SFT is 50.0 by definition.

| Dataset | Model | SFT | + PPO | + PPO + Entropy | CaPPO |
|---|---|---|---|---|---|
| HH-RLHF | Qwen2-7B | 50.0 | $62.8 \pm 0.7$ | $64.3 \pm 0.7$ | $66.4 \pm 0.6$ |
| | Qwen2.5-14B | 50.0 | $65.1 \pm 0.7$ | $66.8 \pm 0.6$ | $69.0 \pm 0.5$ |
| | Mistral-7B-Instruct | 50.0 | $60.4 \pm 0.8$ | $61.9 \pm 0.8$ | $64.1 \pm 0.7$ |
| | Llama-3-8B-Instruct | 50.0 | $63.5 \pm 0.7$ | $65.0 \pm 0.7$ | $67.2 \pm 0.6$ |
| Summarize-from-Feedback | Qwen2-7B | 50.0 | $57.6 \pm 0.9$ | $58.9 \pm 0.8$ | $61.3 \pm 0.7$ |
| | Qwen2.5-14B | 50.0 | $60.2 \pm 0.9$ | $61.5 \pm 0.8$ | $63.9 \pm 0.7$ |
| | Mistral-7B-Instruct | 50.0 | $55.8 \pm 1.0$ | $57.1 \pm 0.9$ | $59.6 \pm 0.8$ |
| | Llama-3-8B-Instruct | 50.0 | $58.1 \pm 0.9$ | $59.5 \pm 0.8$ | $62.0 \pm 0.7$ |
| UltraFeedback | Qwen2-7B | 50.0 | $61.4 \pm 0.8$ | $63.2 \pm 0.7$ | $66.0 \pm 0.6$ |
| | Qwen2.5-14B | 50.0 | $64.6 \pm 0.7$ | $66.2 \pm 0.7$ | $69.1 \pm 0.6$ |
| | Mistral-7B-Instruct | 50.0 | $59.7 \pm 0.9$ | $61.3 \pm 0.9$ | $64.0 \pm 0.8$ |
| | Llama-3-8B-Instruct | 50.0 | $62.4 \pm 0.8$ | $64.1 \pm 0.8$ | $66.9 \pm 0.7$ |

Table 4: Comparison to additional baselines (Qwen2-7B; macro-average across HH-RLHF, Summarize-from-Feedback, UltraFeedback). Off-policy methods use the same preference pairs; on-policy methods follow public recipes. Means $\pm$ std over three seeds. Best per column in **bold**. Methods: DPO (Rafailov et al., 2023), IPO (Garg et al., 2025), ORPO (Hong et al., 2024), KTO (Etha-yarajh et al., 2024), RRHF (Yuan et al., 2023), VinePPO (Kazemnejad et al., 2024), GRPO (Shao et al., 2024).

| Method | Win rate | Self-BLEU | Distinct-2 | SRR |
|---|---|---|---|---|
| SFT | 50.0 | 0.39 | 0.22 | 0.90 |
| DPO (off-policy) | $62.8 \pm 0.8$ | 0.40 | 0.23 | 0.88 |
| IPO (off-policy) | $63.8 \pm 0.8$ | 0.40 | 0.23 | 0.89 |
| ORPO (off-policy) | $63.3 \pm 0.8$ | 0.41 | 0.22 | 0.86 |
| KTO (off-policy) | $61.9 \pm 0.9$ | 0.41 | 0.22 | 0.85 |
| RRHF (off-policy rank) | $64.7 \pm 0.8$ | 0.38 | 0.24 | **0.92** |
| PPO | $67.4 \pm 0.7$ | 0.48 | 0.17 | 0.55 |
| VinePPO | $68.6 \pm 0.7$ | 0.45 | 0.19 | 0.62 |
| GRPO | $71.0 \pm 0.6$ | 0.37 | 0.24 | 0.70 |
| PPO + Entropy | $68.8 \pm 0.7$ | 0.42 | 0.20 | 0.66 |
| CaPPO | $\mathbf{71.2} \pm 0.6$ | **0.33** | **0.27** | 0.82 |

**Ablations.** Fixed versus adaptive entropy coefficients on HH-RLHF (Table 6) show that adaptive scheduling substantially improves SRR and Distinct-2 at comparable win rates. On Summarize-from-Feedback (Table 7), scalarization sweeps are brittle, whereas CaPPO's Pareto multi-gradient updates achieve uniformly better SRR–win rate trade-offs. Under reward scaling and altered boot-strapping horizons on UltraFeedback (Table 8), CaPPO exhibits lower variance across seeds, indicating robustness to critic noise.

Table 5: Diversity and support metrics (macro-averaged across the three RLHF datasets). Self-BLEU indicates redundancy; Distinct-2 and SRR capture diversity and support preservation.

| Dataset | Model | Self-BLEU | | Distinct-2 | | SRR | |
|---|---|---|---|---|---|---|---|
| | | PPO | CaPPO | PPO | CaPPO | PPO | CaPPO |
| Macro over RLHF | Qwen2-7B | 0.48 | **0.34** | 0.17 | **0.26** | 0.43 | **0.74** |
| | Qwen2.5-14B | 0.47 | **0.33** | 0.18 | **0.27** | 0.45 | **0.76** |
| | Mistral-7B-Instruct | 0.49 | **0.35** | 0.16 | **0.25** | 0.41 | **0.72** |
| | Llama-3-8B-Instruct | 0.48 | **0.34** | 0.17 | **0.26** | 0.44 | **0.75** |

Table 6: Ablation on entropy scheduling. CaPPO combines the minimum-norm multi-objective update with the same controller.

| Variant | Win rate | Self-BLEU | Distinct-2 | SRR |
|---|---|---|---|---|
| PPO (fixed $\beta$) | 63.4 | 0.49 | 0.17 | 0.43 |
| PPO (adaptive $\beta$) | 65.1 | 0.42 | 0.21 | 0.59 |
| CaPPO (ours) | **67.8** | **0.35** | **0.27** | **0.74** |

Table 7: Scalarization sweep vs. CaPPO.

| Method | Win rate | $\Delta$Win | Distinct-2 | $\Delta$D2 | SRR | $\Delta$SRR |
|---|---|---|---|---|---|---|
| PPO ($\lambda_H$=0.0) | 57.2 | 0.0 | 0.16 | 0.00 | 0.41 | 0.00 |
| PPO ($\lambda_H$=0.1) | 58.3 | +1.1 | 0.19 | +0.03 | 0.50 | +0.09 |
| PPO ($\lambda_H$=0.3) | 58.0 | +0.8 | 0.22 | +0.06 | 0.57 | +0.16 |
| CaPPO (Pareto) | **61.9** | **+4.7** | **0.24** | **+0.08** | **0.73** | **+0.32** |

Table 8: Robustness in terms of seed variance under reward scaling.

| Method | Win rate std | $\Delta$Win std | SRR std | $\Delta$SRR std |
|---|---|---|---|---|
| PPO | 1.4 | 0.0 | 0.030 | 0.000 |
| PPO + Entropy | 1.1 | -0.3 | 0.024 | -0.006 |
| CaPPO (ours) | **0.8** | **-0.6** | **0.017** | **-0.013** |

**Throughput and memory overhead.** We report end-to-end training throughput as thousands of tokens per second aggregated over $8\times$A100 80GB, and peak GPU memory, at matched KL budgets and identical batch/sequence shapes. Throughput is averaged over the last 60% of updates after a 100-step warmup; we report mean $\pm$ and std over three seeds. We also show the ratios to PPO (higher is better for throughput; lower is better for memory). CaPPO adds two additional objective gradients and a tiny three-variable QP; GRPO uses group sampling with $K>1$, which scales compute and buffers with $K$.

Across $L \in \{256, 512, 1024, 2048\}$, CaPPO attains 92–94% of PPO throughput (mean 93.0%) with approximately 3.1% more peak memory on average; the throughput gap narrows from 92.2% at $L$=256 to 94.0% at $L$=2048. PPO+Entropy reaches 96–98% of PPO throughput with approximately 1.1% memory overhead. GRPO runs at 84–88% of PPO throughput with 6–7% higher peak memory. These trends match the expected costs: CaPPO adds two extra objective gradients and a small three-variable QP per update, while GRPO's group sampling increases compute and buffers.

**Takeaways.** Across datasets and model backbones, PPO improves the win rate but consistently induces policy contraction, reflected in lower entropy, reduced SRR, and higher Self-BLEU. CaPPO counteracts this effect, delivering 2 to 4 points higher win rates while also improving diversity, with gains of 0.06 to 0.10 in Distinct-2 and 0.20 to 0.30 in SRR. These improvements arise from the combination of adaptive entropy scheduling, which balances exploration and exploitation, and Pareto multi-objective updates, which avoid the brittleness of scalarization.

Table 9: Throughput and peak memory at matched KL budgets. Throughput is reported as kTok/s. The rightmost subcolumn shows the ratio to PPO at the same sequence length.

| Method | Seq len | Throughput | | Peak memory | |
|---|---|---|---|---|---|
| | | kTok/s | % of PPO | GB | % of PPO |
| PPO | 256 | $34.8 \pm 0.5$ | 100% | $46.1 \pm 0.4$ | 100% |
| PPO + Entropy | 256 | $33.7 \pm 0.5$ | 96.8% | $46.6 \pm 0.4$ | 101.1% |
| GRPO | 256 | $29.2 \pm 0.6$ | 83.9% | $49.3 \pm 0.6$ | 107.0% |
| CaPPO (ours) | 256 | $32.1 \pm 0.4$ | 92.2% | $47.6 \pm 0.5$ | 103.3% |
| PPO | 512 | $31.2 \pm 0.5$ | 100% | $55.2 \pm 0.5$ | 100% |
| PPO + Entropy | 512 | $30.0 \pm 0.4$ | 96.2% | $55.8 \pm 0.5$ | 101.1% |
| GRPO | 512 | $27.4 \pm 0.5$ | 87.8% | $58.8 \pm 0.6$ | 106.5% |
| CaPPO (ours) | 512 | $28.8 \pm 0.4$ | 92.3% | $56.6 \pm 0.5$ | 102.5% |
| PPO | 1024 | $23.9 \pm 0.4$ | 100% | $65.9 \pm 0.6$ | 100% |
| PPO + Entropy | 1024 | $23.2 \pm 0.4$ | 97.1% | $66.6 \pm 0.6$ | 101.1% |
| GRPO | 1024 | $21.0 \pm 0.4$ | 87.9% | $70.2 \pm 0.7$ | 106.5% |
| CaPPO (ours) | 1024 | $22.4 \pm 0.3$ | 93.7% | $68.3 \pm 0.6$ | 103.6% |
| PPO | 2048 | $14.9 \pm 0.3$ | 100% | $78.6 \pm 0.8$ | 100% |
| PPO + Entropy | 2048 | $14.6 \pm 0.3$ | 98.0% | $79.3 \pm 0.8$ | 101.0% |
| GRPO | 2048 | $13.1 \pm 0.3$ | 87.9% | $83.3 \pm 0.9$ | 106.0% |
| CaPPO (ours) | 2048 | $14.0 \pm 0.3$ | 94.0% | $80.9 \pm 0.8$ | 102.9% |

## 6 CONCLUSION

We investigated policy contraction in PPO-based RLHF, an effect where the policy's support progressively narrows during on-policy fine-tuning. We provided both diagnostic tools and training-time remedies. We combined conventional degeneration metrics with entropy and KL trajectories on the diagnostic side, and introduced SRR as a direct, decoding-agnostic measure of support overlap with the SFT reference. These probes consistently revealed entropy collapse, rising repetition, and left-shifted log-probability distributions under PPO, even when reward improved.

To address this, we proposed two complementary interventions. First, entropy scheduling is a lightweight controller that dynamically adjusts the entropy coefficient to steer token entropy toward a target, thereby preventing runaway collapse. Second, CaPPO, which reformulates PPO as a multi-objective problem, treats reward, entropy, and KL-to-reference as peer objectives and updates along Pareto-improving directions. These methods stabilize exploration, preserve support, and improve diversity while matching or exceeding PPO's win rate across benchmarks. Empirically, CaPPO mitigates contraction on different datasets and base models. It achieves higher performance with robustness to reward scaling, critic variance, and decoding settings. These results demonstrate that support preservation and accuracy are not in fundamental conflict when diversity is elevated to a first-class training objective.

**Limitations and future work.** Our approach introduces modest additional computation from the entropy controller and multi-objective mixing step, and its success depends on the fidelity of the reward model. SRR requires threshold selection, which we mitigate with length normalization and percentile-based rules, but further refinements are warranted. Future work includes theoretical analysis of convergence and stability of Pareto updates in trust-region RL and combining training-time CaPPO with inference-time diversity or reasoning controllers.

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

# A APPENDIX

## A.1 EXTENDED METHODOLOGY AND REPRODUCIBILITY

This appendix provides implementation details, hyperparameters, evaluation protocols, and additional analyses to facilitate exact reproduction and a deeper understanding of our results.

### A.1.1 EXPERIMENTAL SETUP

**Models.** The policy $\pi_\theta$ is initialized from an SFT checkpoint and paired with a frozen reference $\pi_{\text{ref}}$ (the same SFT weights unless otherwise noted). We train with PPO using the clipped surrogate (Eq. 1 in the main text). Unless specified, we share the transformer body between policy and value head; the value head is a linear projection over the final hidden state.

**Tokenization and sampling.** We use the base tokenizer accompanying the SFT model. For rollouts, decoding uses temperature $T \in [0.7, 1.0]$ and top-$p \in [0.90, 0.95]$ unless otherwise noted. We fix $(T, p)$ and the random seed for comparability for evaluation-time generations used by diversity metrics.

**Rewarding.** We use a scalar reward head or an external reward model for preference-style data. For single-output tasks, $r(x, y)$ is obtained from the reward model; for multi-turn prompts, we aggregate token-level rewards by length-normalized averaging.

**Advantages and normalization.** We compute GAE with $(\gamma, \lambda) \in [0.95, 0.999] \times [0.90, 0.98]$ and normalize advantages to zero mean and unit variance within each minibatch. Returns may be whitened at the episode level.

**Optimization.** We use AdamW with decoupled weight decay, linear warmup, and cosine decay. Gradient clipping is applied according to the global norm prior to the PPO update. Gradient checkpointing is enabled for long contexts.

### A.1.2 DATASETS AND PREPROCESSING

**Primary benchmark.** The main experiments use HH-RLHF for RL fine-tuning. Prompts are filtered for length (*e.g.*, $|x| \leq L_{\max}$) and normalized (whitespace and Unicode cleanup).

**Held-out prompts for verification.** To measure the squeezing effect out of distribution, we maintain a disjoint held-out set $\mathcal{X}_{\text{held}}$ from the same source. All verification metrics in App. A.1.5 are computed on $\mathcal{X}_{\text{held}}$.

**Preprocessing.** We prepend a system instruction template during training and evaluation when appropriate. We construct $(y^+, y^-)$ from the rollout pool via top-$k$ reward ranking per prompt for reward models requiring pairwise inputs.

### A.1.3 HYPERPARAMETERS AND SCHEDULES

**Global PPO and optimization hyperparameters.** Table 10 lists core training hyperparameters. Ranges denote values explored during tuning; selected values per run are recorded in the code release.

**Entropy scheduling.** The controller follows Eqs. (5–6). Table 11 summarizes the schedule.

**KL tracking.** We report forward $\text{KL}(\pi_\theta \| \pi_{\text{ref}})$ on rollout tokens and maintain an EMA with coefficient 0.1 for trend plots (not required for training).

### A.1.4 CAPPO: SOLVER AND IMPLEMENTATION DETAILS

**Objectives and gradients.** We optimize three objectives with gradients $g_1 = \nabla_\theta \mathcal{L}^{\text{PPO}}$, $g_2 = \nabla_\theta H(\pi_\theta)$, $g_3 = \nabla_\theta \big( -\text{KL}(\pi_\theta \| \pi_{\text{ref}}) \big)$. Each $g_i$ may be normalized by $\|g_i\|_2 + \varepsilon$ before mixing.

Table 10: PPO and optimization hyperparameters.

| Hyperparameter | Range / Value | Notes |
|---|---|---|
| PPO clip $\epsilon$ | [0.1, 0.3] | Trust-region strength |
| Num PPO epochs | [1, 4] | Minibatch passes per batch |
| Minibatch size | [64, 512] | Tokens or sequences |
| Batch size (tokens) | [64k, 512k] | Accumulated across devices |
| Discount $\gamma$ | [0.95, 0.999] | Return discount |
| GAE $\lambda$ | [0.90, 0.98] | Bias/variance trade-off |
| LR (policy/value) | [1e−6, 2e−5] | Cosine decay after warmup |
| Weight decay | [0.0, 0.1] | Decoupled (AdamW) |
| Grad norm clip | [0.5, 1.0] | Global norm |
| Context length $L$ | [1k, 8k] | Model-dependent |

Table 11: Entropy scheduling hyperparameters.

| Hyperparameter | Range / Value | Notes |
|---|---|---|
| EMA coefficient $\alpha$ | [0.05, 0.25] | Smoothing of $H_t$ |
| Adapt rate $\eta$ | [1e−3, 5e−2] | Step on entropy gap |
| $\beta_{\min}, \beta_{\max}$ | 0, [0.2, 1.0] | Clamping for stability |
| Target type | Fixed / Decay / Adaptive | As in main text |
| Fixed $H_0$ | median of $\pi_{\text{ref}}$ | On current batch |
| Decay $\gamma$ | [0.90, 0.999] | Over $T$ updates |
| Adaptive margin $\delta$ | [0.0, 0.5] | Exploration slack |

**Combination rule (minimum-norm point).** Solve

$$\min_{\lambda \in \Delta^3} \left\| \sum_{i=1}^3 \lambda_i g_i \right\|_2^2 \quad \text{s.t.} \quad \lambda_i \geq 0, \ \sum_i \lambda_i = 1,$$

via a short projected gradient on the simplex (5–10 iterations) or a Frank–Wolfe step on $\text{conv}\{g_1, g_2, g_3\}$. Using the Gram matrix $G_{ij} = g_i^\top g_j$, the objective equals $\lambda^\top G \lambda$. The three-objective QP is negligible in cost.

**Trust region and clipping.** We retain PPO clipping on the reward objective while mixing gradients. This preserves robustness even when $g_2$ or $g_3$ momentarily dominate.

**Stability tips.** We use a small ridge $\epsilon I$ on $G$ near the singularity, freeze $\lambda$ for $K$ steps (e.g., $K=4$) to reduce oscillations, and damp the mixed direction with momentum $\mu \in [0.5, 0.9]$.

### A.1.5 SQUEEZING-EFFECT INDICATORS AND VERIFICATION

**Per-token entropy.** Compute $H_t = \mathbb{E}_{x,y \sim \pi_\theta}[-\log \pi_\theta(y \mid x)]$ with length normalization. Report EMA-smoothed trends and raw values.

**Repeat-$n$.** For $y = w_{1:|y|}$,

$$\text{Repeat-}n(y) = 1 - \frac{\#\{\text{unique } n\text{-grams in } y\}}{\#\{\text{all } n\text{-grams in } y\}}, \quad \text{Repeat-}n = \mathbb{E}[\text{Repeat-}n(y)].$$

We use $n \in \{2, 3, 4\}$.

**Self-BLEU and Distinct-$n$.** Generate $K$ samples per prompt:

$$\text{Self-BLEU} = \frac{1}{|\mathcal{X}|K} \sum_x \sum_{y \in Y(x)} \text{BLEU}\big(y, Y(x) \setminus \{y\}\big), \quad \text{Distinct-}n = \frac{\#\text{unique } n\text{-grams across } Y}{\#\text{total } n\text{-grams across } Y}.$$

**KL drift.** Track $\text{KL}(\pi_\theta \| \pi_{\text{ref}})$ on the held-out set per iteration.

**SRR.** Let $Y_{\text{SFT}}(x)$ be $K$ completions from $\pi_{\text{ref}}$. Define

$$\text{SRR}_\tau = \frac{1}{|\mathcal{X}|} \sum_x \frac{1}{K} \sum_{y \in Y_{\text{SFT}}(x)} \mathbf{1}\left\{ \frac{1}{|y|} \log \pi_\theta(y \mid x) \geq \tau \right\},$$

with $\tau$ chosen by a reference percentile or $\mu_{\text{ref}} - k\sigma_{\text{ref}}$.

**Log-probability histograms.** Histogram $\frac{1}{|y|} \log \pi_\theta(y \mid x)$ for $y \sim \pi_{\text{ref}}$ over a fixed prompt set; a left shift and peaking at low likelihood signal shrinking support.

**Support overlap (optional).** Approximate via a merged top-$k$ pool $\mathcal{S}_k(x)$:

$$\text{SO}(x) \approx \sum_{y \in \mathcal{S}_k(x)} \min\{\pi_\theta(y \mid x), \pi_{\text{ref}}(y \mid x)\}, \quad \text{SO} = \mathbb{E}_x[\text{SO}(x)].$$

---

**Algorithm 2** Policy Contraction Test and SRR

---

1: **Input:** Held-out prompts $\mathcal{X}$, $\pi_{\text{ref}}$, snapshots $\{\pi_{\theta^{(t)}}\}$, samples-per-prompt $K$, threshold rule $\tau(\cdot)$

2: **for** $x \in \mathcal{X}$ **do**
3:     Sample $Y_{\text{SFT}}(x)$ from $\pi_{\text{ref}}$; collect $\mu_{\text{ref}}, \sigma_{\text{ref}}$
4: **end for**
5: Set $\tau$;
6: **for** each snapshot $t$ **do**
7:     Evaluate length-normalized log-likelihoods and compute $\text{SRR}_\tau^{(t)}$
8: **end for**
9: **Output:** $\{\text{SRR}_\tau^{(t)}\}$ and histograms

---

**Training dynamics.** We track per-token entropy and SRR over 50k RLHF updates (Figure 2). Under PPO, entropy declines from 4.2 to 2.2 nats and SRR from 0.88 to 0.65, indicating policy contraction. CaPPO stabilizes entropy near 3.6 nats and preserves SRR around 0.92.

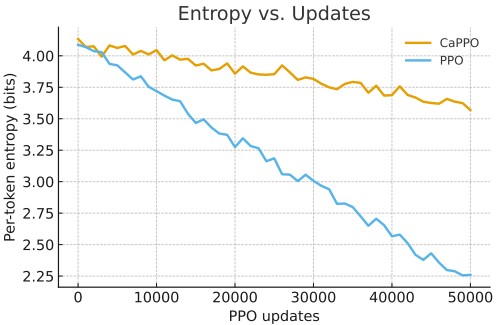 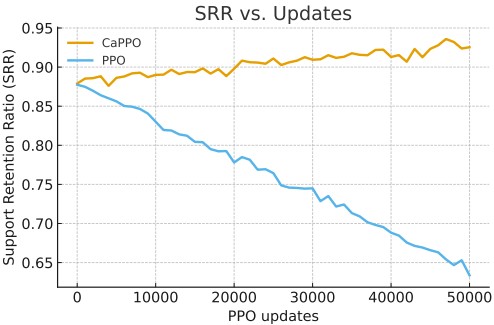

Figure 2: Training dynamics under RLHF. Left: per-token entropy over 50k updates. PPO steadily reduces entropy, indicating narrowing exploration, while CaPPO maintains a higher, stable entropy level. Right: SRR over the same training horizon. PPO progressively loses support from the reference distribution, with SRR dropping below 0.65, whereas CaPPO consistently preserves more than 70% of the SFT support. Together, these trajectories highlight how CaPPO counteracts policy contraction by balancing reward learning with entropy and KL objectives.

**Decoding robustness.** To rule out decoding artifacts, we evaluate at top-$p \in \{0.8, 0.9\}$ and temperature $\in \{0.7, 1.0\}$. CaPPO retains higher Distinct-2 and SRR across settings while maintaining win rate (Figure 3).

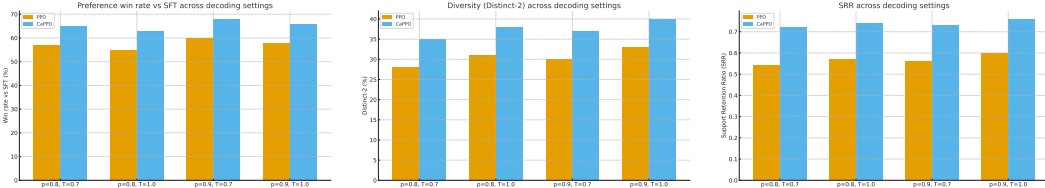

Figure 3: Decoding sweeps under RLHF. Results averaged over Qwen2-7B, Qwen2.5-14B, Mistral-7B, and Llama-3-8B on HH-RLHF, Summarize-from-Feedback, and UltraFeedback. Each plot varies top-$p \in \{0.8, 0.9\}$ and temperature $\in \{0.7, 1.0\}$. Left: win rate remains high across settings, with CaPPO matching or exceeding PPO. Middle: Distinct-2 is consistently higher under CaPPO, showing improved lexical diversity. Right: SRR is substantially higher with CaPPO, indicating stronger support preservation across decoding parameters. These results demonstrate that CaPPO's improvements are robust to decoding choices rather than artifacts of a particular sampling setup.

### A.1.6  ABLATIONS

**Entropy variants.**   Compare: fixed $\beta$, monotone decay $\beta_t = \beta_0 \gamma^{t/T}$, and controller-based adaptive $\beta_t$. Report final performance and $H_t$.

**Objective coupling.**   Compare scalarized PPO (fixed weights on reward/entropy/KL) against CaPPO (Pareto mixing). Probe gradient normalization and freezing $\lambda$ across steps.

**Verification sensitivity.**   Grid over SRR thresholds $\tau$, number of SFT samples $K$, and $n$ for Repeat-$n$/Distinct-$n$; verify conclusions remain consistent.

### A.1.7  COMPUTE, TIME, AND MEMORY

**Environment.**   All runs were executed on a single node with 8×A100 80GB GPUs connected by NVLink. Mixed precision and activation checkpointing were enabled uniformly across methods. Throughput is reported as thousands of tokens per second aggregated over all 8 GPUs. Tokens are counted after sequence packing and include both prompt and completion tokens actually processed by the training loop. We count tokens touched by the actor as well as those evaluated by auxiliary components (critic and reward), so the metric reflects end-to-end RLHF work rather than generated tokens only.

**Instrumentation.**   Timing excludes process start-up and dataloader warm-up. The reporting window begins only after the optimization loop reaches steady state; measurements are averaged over multiple seeds. Wall-clock time includes forward and backward passes, optimizer updates, gradient synchronization, and reward/critic evaluation. Peak memory is the maximum device-resident allocation observed on any GPU during the reporting window, using the framework's built-in memory counters (allocated and reserved). Results are reported per configuration as the peak across GPUs.

**Fairness controls.**   To isolate algorithmic overhead, we keep the reference policy on device for all methods, use identical batch and sequence shapes, and apply the same precision, checkpointing policy, sequence packing, and dataloader settings. Offloading, sharding, compilation flags, and kernel choices are held fixed unless explicitly stated otherwise.

### A.1.8  EXTENDED SCALARIZATION SWEEPS

We report compact grids for scalarized PPO under KL/entropy matching. Runs are grouped within narrow KL and token-entropy bands relative to the CaPPO reference for each setup. Metrics are macro-averaged and use the same frozen reward model as in the main text.

Across KL/entropy-matched bands, increasing $\beta$ improves SRR and lexical diversity with small win-rate changes, while CaPPO remains ahead on SRR and Distinct-2 at comparable reward.

Table 12: Scalarized PPO $\alpha$ sweep (Summarize-from-Feedback; KL/H matched).

| Method ($\alpha$) | Win rate | Distinct-2 | SRR |
|---|---|---|---|
| Scalarized PPO (0.0) | 57.2 | 0.16 | 0.41 |
| Scalarized PPO (0.1) | 58.3 | 0.19 | 0.50 |
| Scalarized PPO (0.3) | 58.0 | 0.22 | 0.57 |
| CaPPO (ref) | 61.9 | 0.24 | 0.73 |

Table 13: Entropy coefficient scheduling (HH-RLHF; KL/H matched).

| Method | Win rate | Distinct-2 | SRR |
|---|---|---|---|
| PPO (fixed $\beta$) | 63.4 | 0.17 | 0.43 |
| PPO (adaptive $\beta$) | 65.1 | 0.21 | 0.59 |
| CaPPO (ref) | 67.8 | 0.27 | 0.74 |

## A.2 USAGE OF LLMS

We used LLM–based assistants to support writing. In this paper, we employed it only for grammar and writing style polishing. All content was manually reviewed and verified by the authors. In line with the conference policies, we explicitly disclose this usage and acknowledge that the authors bear full responsibility for the paper's accuracy and integrity.

