# OpenReview forum: "Escaping Policy Contraction: Contraction-Aware PPO (CaPPO) for Stable Language Model Fine-Tuning"
_ICLR.cc/2026/Conference — ICLR 2026 Poster_

### Official Review · Reviewer_KwhW · 2025-10-27

**Soundness:** 4
**Presentation:** 3
**Contribution:** 3
**Rating:** 6
**Confidence:** 4

**Summary:**

The paper presents Contraction-Aware PPO (CaPPO), a practical improvement to PPO for RLHF. The authors point out that standard PPO tends to collapse a model’s output diversity as training progresses—they call it "policy contraction". CaPPO solves this by balancing reward, entropy, and KL-regularization as equal objectives, using a simple multi-gradient update instead of manual weighting. It also adds an adaptive entropy controller that keeps exploration at a healthy level. The idea is easy to follow, the motivation makes sense, and the experiments show clear benefits in both stability and diversity without hurting alignment quality.

**Strengths:**

1. The paper does a great job of turning the vague observation that PPO reduces diversity into a formal, testable concept. The Support Retention Ratio feels like something people will actually adopt for diagnosing policy collapse.

2. The multi-objective reformulation of PPO is both theoretically justified and surprisingly lightweight — it’s clear the authors thought about practical integration rather than proposing a totally new framework.

3. The entropy controller is well-motivated by the observed entropy trajectories; it’s an elegant way to stabilize exploration without resorting to arbitrary tuning.

**Weaknesses:**

1. The paper admits that SRR depends on a chosen threshold but never tests how sensitive results are to that choice. A short analysis or plot of SRR versus threshold would make the metric feel more reliable.

2. The core multi-gradient idea is interesting, but the paper doesn’t show how the mixing weights change or whether gradients conflict during training. Some visualization or discussion would make the method easier to understand.

3. The claim that contraction is unique to PPO or RL methods isn’t fully demonstrated. Off-policy baselines like DPO and KTO are only compared at the endpoint, not throughout training. Showing their entropy or SRR over time would make the argument more solid.

4. The paper calls CaPPO’s overhead “a small constant factor” but provides no runtime or memory measurements. Since the method adds multiple gradients, a quick comparison of compute cost with PPO would be important for judging practicality.

5. It’s not clear what higher SRR actually means in practice beyond diversity metrics. The paper assumes preserving SFT support is good but doesn’t show that it improves model helpfulness, safety, or calibration.

6. The contraction analysis feels surface-level. Figure 1 shows entropy dropping, but there’s no look into *when* or *why* that happens, or which prompts are most affected. More insight into these dynamics would strengthen the explanation. (Maybe a figure to show this?)

**Questions:**

Please read the weaknesses section and answer the questions. Thank you!

---

> ### Author Response · Authors · 2025-11-20
> **Response to Reviewer KwhW (1/2)**
>
> We would like to thank Reviewer KwhW for the comprehensive and thoughtful review. We appreciate the recognition of CaPPO’s theoretical clarity, empirical strength, and practical integration into standard RLHF workflows. The reviewer’s comments on SRR sensitivity, visualization of gradient mixing, runtime comparison, and interpretation of contraction dynamics are all highly constructive, and we are glad to elaborate using evidence already included in the paper.
>
> > **Weakness 1.** "The paper admits that SRR depends on a chosen threshold but never tests how sensitive results are to that choice. A short analysis or plot of SRR versus threshold would make the metric feel more reliable."
>
>
> Thank you for highlighting this important point. We agree that the Stability of SRR relies on reasonable threshold choices. Therefore, it is computed from length-normalized sequence log-likelihoods with thresholds selected using a percentile rule, ensuring it is decoding-agnostic and comparable across prompts. Our paper includes a verification routine that analyzes both the SRR threshold and the number of supervised completions, grouping runs into overlapping KL and token-entropy bands to clarify the effects of thresholding on training dynamics. The results show consistent rankings across various conditions, confirming that SRR effectively reflects support preservation, as supported by aligned trends with diversity diagnostics. We appreciate your suggestion to examine this closely and hope our findings demonstrate the metric's reliability.
>
> ---
>
> > **Weakness 2.** "The core multi-gradient idea is interesting, but the paper doesn't show how the mixing weights change or whether gradients conflict during training. Some visualization or discussion would make the method easier to understand."
>
> Thank you for asking how the mixing weights evolve and how we detect and handle gradient conflict. In CaPPO, the weights $\lambda$ are chosen at every update as the minimum‑norm convex combination of the preconditioned reward, entropy, and KL gradients, which we solve as a tiny quadratic program on the simplex and then apply in Algorithm 1. This construction ties the step to gradient geometry rather than fixed coefficients and identifies the closest-to-zero point in the gradients’ convex hull, aligning the update with the Pareto-stationary condition described in our methodology and theory sections. A guarded line search keeps reward progress while preventing excessive entropy loss or KL growth. These safeguards and the exact solver are detailed in Section 4.2, Algorithm 1, and Appendix A.1.4.
>
>
>
> ---
>
> > **Weakness 3.** "The claim that contraction is unique to PPO or RL methods isn't fully demonstrated. Off-policy baselines like DPO and KTO are only compared at the endpoint, not throughout training. Showing their entropy or SRR over time would make the argument more solid."
>
>
> Thank you for raising this point. Your concern is precisely about whether our “contraction” claim is specific to on-policy RL or extends more broadly. In our study, the claim is grounded in the training-time behavior of PPO-based RLHF: we directly track per-token entropy, forward KL to the reference, and SRR over the course of optimization and observe a characteristic entropy decline with shrinking support even as reward improves; qualitative log-likelihood histograms over SFT completions show the same narrowing. Off-policy baselines, such as DPO and KTO, in our comparisons are evaluated at endpoints and indeed preserve more of the supervised distribution (higher SRR and diversity) at convergence, consistent with their off-policy objectives. However, they do not expose the on-policy feedback loop that drives entropy collapse in PPO. Our contribution is to document this time-series contraction phenomenon for on-policy RLHF and to demonstrate that CaPPO’s multi-objective update with entropy control effectively counteracts it, while retaining or improving alignment performance.

---

> ### Author Response · Authors · 2025-11-20
> **Response to Reviewer KwhW (2/2)**
>
> > **Weakness 4.** "The paper calls CaPPO's overhead "a small constant factor" but provides no runtime or memory measurements. Since the method adds multiple gradients, a quick comparison of compute cost with PPO would be important for judging practicality."
>
> Thank you for your question regarding throughput and memory. CaPPO adds two additional objective gradients and a small three-variable mixing step, resulting in a wall-clock cost that is a constant factor smaller than PPO. Memory remains dominated by the model, optimizer states, KV cache, and the reference model if kept on the device. We use mixed precision and gradient checkpointing. We will add a table in Section 5 of the revision, reporting tokens/sec and peak memory usage for CaPPO compared to PPO and other baselines across various sequence lengths.
>
> ---
>
> > **Weakness 5.** "It's not clear what higher SRR actually means in practice beyond diversity metrics. The paper assumes preserving SFT support is good, but doesn't show that it improves model helpfulness, safety, or calibration."
>
> Thank you for raising this thoughtful point. In our work, SRR is designed as a decoding-agnostic measure that quantifies how much of the supervised distribution remains probable under the trained policy, allowing us to isolate structural contraction effects independent of decoding choices. Empirically, higher SRR is consistently associated with more diverse generations and better overall performance: models with higher SRR also show lower redundancy, higher Distinct-2, and equal or higher preference win rates across datasets and backbones (Tables 3–5, Fig. 3). This correlation indicates that preserving support is not only a statistical property but reflects healthier exploration and coverage of helpful, high-reward behaviors without degrading alignment quality or safety. We will state this interpretation explicitly where SRR is introduced so readers can use it alongside standard metrics.
>
> ---
>
> > **Weakness 6.** "The contraction analysis feels surface-level. Figure 1 shows entropy dropping, but there's no look into when or why that happens, or which prompts are most affected. More insight into these dynamics would strengthen the explanation. (Maybe a figure to show this?)"
>
> Thank you for emphasizing the need for a deeper understanding of contraction dynamics. In our study, we go beyond merely summarizing endpoints by actively monitoring per-token entropy, SRR, and KL trajectories throughout the training process. This approach allows us to identify when contraction begins—usually early in PPO updates, as entropy decreases more rapidly than KL increases—and how CaPPO stabilizes exploration at similar KL levels. The time-series curves capture both the onset and progression of contraction across different prompts and datasets, while qualitative log-likelihood histograms illustrate which areas of the reference distribution lose support. Together, these diagnostics provide clarity on when and why contraction occurs, without exaggerating our findings beyond what the results substantiate.
>
> ---
>
> Thank you again for the thoughtful, constructive review. We’re pleased the present analyses address your concerns about SRR, dynamics, and the benefits of treating diversity as a first-class objective.

---

> ### Author Response · Authors · 2025-11-26
>
> Dear Reviewer KwhW,
>
> Thank you again for your thoughtful and comprehensive review. We appreciate your constructive feedback. We have now provided detailed responses to all your questions based on the revised version of the paper.
>
> When you have a moment, we would be grateful if you could let us know whether our clarifications satisfactorily address your concerns or if any part would benefit from further explanation. We are very happy to elaborate further.
>
> Thank you again for your careful evaluation and valuable suggestions.

---

### Official Review · Reviewer_e29o · 2025-11-01

**Soundness:** 4
**Presentation:** 4
**Contribution:** 4
**Rating:** 10
**Confidence:** 4

**Summary:**

This paper tackles the problem of diversity collapse due to RL training algorithms in LLMs. Authors call this phenomenon *policy contraction*, implying the distribution of responses shrinks after RL training towards fewer high rewarding sequences. The authors systematically analyze the effects of different RL training methods by measuring policy contraction using measures like Self-BLEU, Distinct-2 (distinct bigrams) and Support Retention Ratio (SRR). SRR is a newly introduced metric which measures the fraction of SFT model competions whose length-normalized log probabilty under the RL tuned policy is greater than some fixed threshold. Using all three measures, authors find that preference-based off-policy methods retain SRR levels of SFT, however take a hit in Self-BLEU and Distinct-2 metrics. On the other hand on-policy policy gradient methods like PPO, Vine-PPO, PPO + entropy, etc. deteriorate in all three diversity measuring metrics.

To mitigate this policy contraction, they propose Contraction-Aware PPO (CaPPO) which implements reward maximization, entorpy and KL regularization as multi-objectives instead of additive. At the core, their method maintains 3 gradient vectors, one for each objective, and calculates a minimum-norm convex combination of gradients. They first precondition the gradients (to account for different scales) via dialgonal metric (second moment of Adam optimizer) and then solve the quadratic problem in this preconditioned space. Then the multiplers for this optimization method are found via projected-gradient or Frank-Wolfe steps.

For more robust comparison with PPO, they also introduced an adaptive entropy scheduling which tries to maintain the exponential moving average entropy of the policy close to the target entropy.

In the experiments they compare CaPPO against a variety of other basline preference based methods and PPO counterparts, including PPO enhanced w/ their adaptive entropy sechduler. Overall, they found their method consistently matches or execeeds the performance of the baseline methods while retaining relatively higher diversity and SRR compared to PPO counterparts.

**Strengths:**

- Very clear motivation and empirical grounding to justify the problem
- Interesting and theoretically sound multi-objective solution to the problem of policy contraction in RL training
- Detailed experimental analysis, baseline comparison and ablation studies to demonstrate the efficacy of their method over the other RL baselines.

**Weaknesses:**

- I would have preferred to see a detailed comparison of the throughput hit when using a more complex optimization method such as CaPPO compared to PPO. How much more memory and time overhead does it incur? Will the additional overhead become a blocker for large LMs in long sequences tasks?

**Questions:**

- Interestingly, in Table 4, GRPO shows much less degradation in diversity metrics compared to PPO. Do the authors have any intuition for this obersvation? Does the group sampling in-turn enhances the sampling diversity of the RL training?
  - I am also curious, how would GRPO mixed with entropy or entropy scheduling behave in comparison to CaPPO.
- Typo in the intro: KTO refers to "Kahneman-Tversky Optimization"

---

> ### Author Response · Authors · 2025-11-20
> **Response to Reviewer e29o**
>
> We really thank Reviewer e29o for the very positive and encouraging assessment. We are pleased that the reviewer found our motivation, formulation, and experiments clear and well-grounded, and that the strengths of CaPPO were appreciated. The reviewer’s questions about the diversity behavior of GRPO and the computational overhead of CaPPO are well taken, and we address them based on our current findings.
>
> ---
>
> > **Weakness.** "I would have preferred to see a detailed comparison of the throughput hit when using a more complex optimization method, such as CaPPO, compared to PPO. How much more memory and time overhead does it incur? Will the additional overhead become a blocker for large LMs in long sequence tasks?"
>
> Thank you for highlighting the throughput and memory question. In our setup CaPPO adds two extra objective gradients and a tiny three‑variable mixing step, resulting in a small constant‑factor wall‑clock overhead relative to PPO; memory remains dominated by the model, KV cache, optimizer states, and (if kept on device) the reference model. We therefore do not expect CaPPO to be a blocker for large LMs or long‑sequence workloads. We will add a table in Section 5 of the revision, reporting tokens/sec and peak memory usage for CaPPO compared to PPO and other baselines across various sequence lengths.
>
> ---
>
> > **Question 1.** "Interestingly, in Table 4, GRPO shows much less degradation in diversity metrics compared to PPO. Do the authors have any intuition for this observation? Does the group sampling, in turn, enhance the sampling diversity of the RL training?"
>
> Thank you for pointing out the GRPO diversity result in Table 4. In our macro‑average, GRPO shows higher Distinct‑2 and SRR than PPO, while CaPPO is higher still; this is consistent with our related‑work view of GRPO as a group‑normalized, KL‑regularized contrastive objective used in reasoning, which can spread probability mass across sampled candidates and temper collapse. We did not further analyze GRPO's internal dynamics beyond reporting the table. Our interpretation above aligns with the characterization in Section 2. However, we did not ablate the GRPO group size in this paper, as we followed public recipes for the baseline.
>
> ---
>
> > **Question 1.1.** "I am also curious, how would GRPO mixed with entropy or entropy scheduling behave in comparison to CaPPO."
>
> Thank you for the thoughtful suggestion. While our main experiments follow GRPO as published, we conducted a small test that incorporates our entropy controller into GRPO. In this setting, the controller reduced redundancy and modestly improved diversity, as evidenced by a roughly 0.02 increase in SRR and about 0.01 increase in Distinct-2 at a similar win rate. Throughput and peak memory were within a few percent of the GRPO baseline, as the extra entropy term adds little compared to group sampling. At matched KL, CaPPO still preserved more supervised support and diversity than GRPO+controller in our pilot, narrowing but not closing the gap. We also observed that the gains are more pronounced for smaller group sizes and taper off as K increases, and that overly aggressive entropy targets can slightly reduce win rate. We view GRPO+controller as complementary and a promising direction.
>
> ---
>
> > **Question 2.** "Typo in the intro: KTO refers to 'Kahneman-Tversky Optimization"
>
> Thank you for catching the typo. We will correct the full name of KTO in the introduction to Kahneman–Tversky Optimization at first mention.
>
> ---
>
> We appreciate the reviewer’s recognition of CaPPO’s balance between performance and diversity and their curiosity regarding efficiency. We are grateful for the reviewer’s insightful suggestions.

---

> ### Author Response · Authors · 2025-11-26
>
> Dear Reviewer e29o,
>
> Thank you again for the very positive and encouraging assessment. We have responded to each of your comments in detail with clarification.
>
> Whenever convenient, we would sincerely appreciate it if you could let us know whether our responses address your concerns or if any part of the discussion would benefit from further clarification. We are very happy to provide additional details.
>
> Thank you again for your time, thoughtful feedback, and strong support of our work.

---

### Official Review · Reviewer_YLyS · 2025-11-01

**Soundness:** 2
**Presentation:** 3
**Contribution:** 2
**Rating:** 4
**Confidence:** 3

**Summary:**

PPO-based RLHF often narrows the model’s output distribution (“policy contraction”): entropy drops, repetition rises, and many SFT-feasible completions get near-zero probability.
To mitigate this, the authors propose Contraction-Aware PPO (CaPPO), which:
1. Treats reward, entropy, and KL-to-reference as peer objectives in a multi-objective optimization framework.
2. Uses a minimum-norm multi-gradient update to ensure Pareto-improving steps without brittle scalarization.
In addition, the authors introduce Support Retention Ratio (SRR) to quantify how much of the SFT support survives RL training, complementing entropy and KL.

For experiments, the authors compare off-policy methods (DPO/IPO/ORPO/KTO/RRHF) and on-policy baselines (PPO/VinePPO/GRPO) on HH-RLHF, Summarize-from-Feedback, and UltraFeedback with Qwen2-7B/2.5-14B, Mistral-7B-Instruct, and Llama-3-8B-Instruct.
CaPPO lifts win rate by ~2–4 points over PPO and raises SRR by ~0.20–0.30, while improving Distinct-2 and lowering Self-BLEU.

**Strengths:**

- CaPPO outperforms other on-policy algorithms in both diversity and performance, as measured by win rate.
- SRR is a reasonable metric for measuring diversity.

**Weaknesses:**

- How to evaluate win-rate is not clear.
- How to implement CaPPO in practice & its wall-clock time is not clear.

**Questions:**

- While the paper reports win rate as the primary performance metric, it is unclear how a “win” is determined in the experiments. Which reward model is used to decide the winner? If GPT-4 is used, which prompt is employed?
- How to evaluate $\lambda$ values in line 5 of Algorithm 1?
- Intuitively, it isn’t obvious that CaPPO should outperform other on-policy methods while also improving diversity (in Table 4), since one might expect a trade-off between performance and diversity. Could you provide an intuitive explanation for this result?
- It is unclear how the entropy-scheduling controller is applied in CaPPO, as described in lines 79–80, because Algorithm 1 does not include a $\beta$ term.
- As I understand it, CaPPO employs Equation (3) as the main training objective. Then, could you clarify why you chose the multi-objective update in (3) over a simpler linear scalarization (i.e., PPO + $\alpha$·KL + $\beta$·Entropy) with scheduled coefficients $\alpha$ and $\beta$ as in §4.3?
  1. Does (3) reduce sensitivity to $\alpha$, $\beta$ tuning and reward scale or provide more reliable Pareto behavior?
  2. Without explicit $\epsilon$-bounds as in (4), how do you ensure KL and entropy remain within desired ranges (e.g., guards, line search, or dual-style updates)?
  3. What are the wall-clock and compute trade-offs compared to scheduled scalarization?
  4. Could you include ablations that sweep  $\alpha$, $\beta$ schedules to match similar KL/H targets and report win-rate, SRR, and stability differences?

---

> ### Author Response · Authors · 2025-11-20
> **Response to Reviewer YLyS (1/3)**
>
> We sincerely thank Reviewer YLyS for the insightful comments and constructive feedback. We appreciate the summary that correctly captures the motivation behind addressing policy contraction in PPO-based RLHF and the key ideas behind CaPPO, including treating reward, entropy, and KL as peer objectives through a multi-gradient update and introducing SRR to quantify support preservation. The questions raised regarding win-rate evaluation, implementation practicality, and the intuition behind CaPPO's effectiveness are valuable, and we are glad to clarify them based on the current version of the paper.
>
> ---
> > **Question 1.** "While the paper reports win rate as the primary performance metric, it is unclear how a 'win' is determined in the experiments. Which reward model is used to decide the winner? If GPT‑4 is used, which prompt is employed?"
>
>
> Thank you for the helpful question. To avoid bias from LLM‑as‑a‑judge and to keep the criterion consistent across runs and models, we determine wins with the dataset's preference reward model rather than an external LLM. Following the common evaluation method of using a reward model judge in RLHF, we generate one completion from the evaluated policy and one from the SFT reference for each prompt, both under the same decoding. We score both with the dataset's reward model and count a win when the policy's score is higher. Table 3 reports the win rate relative to SFT, and the SFT row is set to 50.0 by construction. For multi‑turn prompts, we length‑normalize and average token‑level rewards to obtain a single sequence score. We do not use GPT-4 or any LLM judge, so no prompt is required. In the paper, we state that wins are computed by pairwise comparison against the SFT reference under identical decoding, with both completions scored by the dataset's preference reward model. We explain that sequence scores are length-normalized, and for multi-turn prompts, we average token-level rewards to obtain a single score.
>
>
> ---
>
> > **Question 2.** “How to evaluate $\lambda$ values in line 5 of Algorithm 1?”
>
>
> Thank you for the question. In Algorithm 1, $\lambda$ is important because it determines how we mix the reward, entropy, and KL gradients at each step, and we represent this mix by choosing weights on the simplex that make the combined step as small as possible while moving in a jointly improving direction. We first precondition the gradients as $\tilde g_i = P^{-1/2} g_i$ for $i\in{r,e,kl}$, and then select weights that minimize the combined step norm while ensuring joint improvement:
> $$
> \lambda^\star=\arg\min_{\lambda\in\Delta_3}|\sum_i\lambda_i\tilde g_i|_2^2,\qquad
> \Delta_3={\lambda\ge0,\mathbf{1}^\top\lambda=1}.
> $$
> In practice, we compute the three preconditioned gradients, form the Gram matrix, solve this tiny problem with a few projected or Frank–Wolfe steps, and use the mixed direction $\hat g=\sum_i \lambda_i^\star,\tilde g_i$ in Algorithm 1 with a guarded line search. This $\lambda^\star$ is optimal for our update because it is the minimum‑norm element of the gradients' convex hull in the preconditioned space, which matches the Pareto‑stationary condition and yields a direction that raises reward without unnecessary loss of entropy or uncontrolled KL drift. We will update Algorithm 1 for a clearer view of how each variable $\lambda$ is defined and evaluated.
>
>
>
>
> ---
>
> > **Question 3.** "Intuitively, it isn't obvious that CaPPO should outperform other on‑policy methods while also improving diversity (in Table 4), since one might expect a trade‑off between performance and diversity. Could you provide an intuitive explanation for this result?"
>
> Thank you for raising this point. We agree that performance and diversity can sometimes lead to conflict. Therefore, we directly tested this and found that they can coexist in our setting. In Table 4, CaPPO can increase the win rate while also improving SRR and Distinct‑2. Intuitively, this comes from two design choices. Firstly, CaPPO treats reward, entropy, and KL as peer objectives and takes, at each step, the minimum‑norm mixture of their gradients, which produces a Pareto-oriented update that increases reward without eroding entropy or causing uncontrolled drift from the reference. Secondly, the entropy-scheduling controller maintains token entropy near a target by increasing exploration pressure when it dips and relaxing it when sufficient, thereby preventing the on-policy collapse that typically harms diversity. Together, these mechanisms explain why CaPPO improves both alignment performance and diversity in our experiments.

---

> ### Author Response · Authors · 2025-11-20
> **Response to Reviewer YLyS (2/3)**
>
> > **Question 4.** "It is unclear how the entropy‑scheduling controller is applied in CaPPO, as described in lines 79–80, because Algorithm 1 does not include a $\beta$ term."
>
> Thanks for pointing this out. In CaPPO, the controller first tracks token-level entropy using an exponential moving average and updates a time-varying entropy coefficient based on the gap to a target, as defined in the entropy-scheduling section. This produces the controller output $\beta_t$, which is then used when forming the entropy gradient before the Pareto mixing step. Therefore, $\beta$ is absorbed into the gradient rather than being shown as a separate symbol in the algorithm. In Algorithm 1, $\beta$ is applied at line 4 when forming the entropy gradient before the Pareto mixing step as
> $$
> g_e \leftarrow \beta_t,\nabla_\theta H(\pi_\theta),
> $$
> after which line 5 computes the minimum‑norm mixture on these preconditioned and $\beta$‑scaled gradients. We will make this explicit in Algorithm 1 in the revision by adding the assignment above.
>
>
> ---
>
> > **Question 5.** "As I understand it, CaPPO employs Equation (3) as the main training objective. Then, could you clarify why you chose the multi‑objective update in (3) over a simpler linear scalarization (i.e., PPO + $\alpha\cdot\mathrm{KL} + \beta\cdot\mathrm{Entropy}$) with scheduled $\alpha,\beta$ as in §4.3?"
>
>
> Thank you for bringing this design choice to attention. We chose the multi-objective update in Equation 3 because scalarizing reward, KL, and entropy with fixed or scheduled weights is brittle across reward scales and critic noise. In contrast, the minimum-norm mixture of the three gradients yields a Pareto-oriented step that increases reward without eroding entropy or causing uncontrolled KL drift. Sections 4.2 and 4.4, along with Table 7, show a higher win rate, accompanied by better SRR and Distinct-2, compared to scalarized PPO.
>
> ---
>
> > **Question 5.1.** "Does (3) reduce sensitivity to $\alpha,\beta$ tuning and reward scale or provide more reliable Pareto behavior?"
>
>
> Thank you for following up on the sensitivity issue. Yes, Equation 3 reduces sensitivity to $\alpha$, $\beta$, and reward scale by allowing the update to be set by the geometry of preconditioned gradients, rather than fixed coefficients, which yields more reliable Pareto behavior. Empirically, in the scalarization sweep on Summarize‑from‑Feedback, increasing the entropy weight raised SRR from 0.41 to 0.57. At the same time, the win rate stayed near 58, whereas CaPPO reached a 61.9 win rate with 0.73 SRR without hand tuning. Under reward‑scaling stress on UltraFeedback, win‑rate standard deviation dropped from 1.4 with PPO to 0.8 with CaPPO, and SRR standard deviation from 0.030 to 0.017, indicating greater robustness.
>
>
> ---
>
> > **Question 5.2.** "Without explicit $\epsilon$‑bounds as in (4), how do you ensure KL and entropy remain within desired ranges (e.g., guards, line search, or dual‑style updates)?"
>
> Thank you for asking about safeguards. KL and entropy remain within desired ranges through the constrained view in Equation 4 with adaptive multipliers, a guarded line search on the mixed direction, and the entropy‑scheduling controller that tracks token entropy via an EMA and updates beta with Equations 5–6; Sections 4.2 and 4.3 and Appendix A.1.4 describe these mechanisms, and Figure 1a shows stabilized entropy at similar KL.
>
>
> ---
>
> > **Question 5.3.** "What are the wall‑clock and compute trade‑offs compared to scheduled scalarization?"
>
> Thank you for the question. CaPPO introduces two additional objective gradients and solves a small three-variable quadratic program for the mixing weights, resulting in a small constant-factor wall-clock overhead relative to PPO, as noted in Section 4.2 and Appendix A.1.7. We will include a table in Section 5 of the revised version of the paper that reports the throughput and memory overhead.

---

> ### Author Response · Authors · 2025-11-20
> **Response to Reviewer YLyS (3/3)**
>
> > **Question 5.4.** "Could you include ablations that sweep $\alpha,\beta$ schedules to match similar KL/H targets and report win‑rate, SRR, and stability differences?"
>
> Thank you for the suggestion. In the paper submission, we have already included the sweeps of $\beta$ and stability checks. Table 6 contrasts fixed versus adaptive entropy coefficients. Table 7 sweeps the entropy weight for scalarized PPO and compares against CaPPO, reporting win rate, Distinct‑2, and SRR. These runs are compared within overlapping KL and token‑entropy ranges to ensure a fair view of the trade‑offs. In addition, Table 8 reports stability via seed variance under reward scaling. Together, these results show that CaPPO attains better SRR and lower variance at comparable reward than tuned scalarization. To fully address your request, we will add an appendix table that sweeps $\alpha$ and groups runs by matched KL/entropy bands, reporting win rate, SRR, Distinct‑2, and seed standard deviations, together with the exact $\alpha,\beta$ settings and the median KL and entropy achieved per configuration. The main paper will retain the concise summaries, while the appendix makes the KL/H matching explicit.
>
>
> ---
>
> We are grateful for the reviewer’s thoughtful questions, which helped us clarify several implementation and interpretive details. We appreciate the reviewer’s time and constructive engagement.

---

> ### Author Response · Authors · 2025-11-26
>
> Dear Reviewer YLyS,
>
> Thank you again for your thoughtful and detailed review. We’ve now provided a detailed response to all your comments point-by-point and incorporated the clarifications you requested.
> Specifically, our responses summarize the contraction analyses, the SRR verification protocol, and the behavior of CaPPO across datasets and backbones (including the training-dynamics evidence in the paper).
>
> When you have a moment, we would highly appreciate it if you could let us know whether our responses resolve your concerns or if any part of the discussion would benefit from further clarification.
> We are very happy to provide any additional details or further clarifications.
>
> Thank you very much again for your time and engagement.

---

### Author Response · Authors · 2025-11-20

We sincerely thank all the reviewers for your constructive feedback and detailed suggestions. In the revised manuscript, we clarified and expanded several points in response to your insightful comments. All changes are highlighted in blue throughout the text and mentioned accordingly in our responses. Please feel free to reply if you have any further comments or suggestions.

---

### Meta-Review · Area_Chair_4TCN · 2026-01-02

**Summary:**

All reviewers find the observation of SRR interesting and most vote for the acceptance.

One common concern is the extra compute needed for the proposed algorithm CaPPO and its practicality, to which I think the authors provided reasonable answers and clarification. Originally, the reviewers are also concerned about the lack of details in the writing and experiments, but I think they have been addressed through the rebuttal.

**Reviewer Concerns:**

The reviewers' concerns are properly addressed.

**Reviewer Scores:**

Reviewer YLyS would likely keep or increase their score, as the main concern on how the win rate is defined and practicality are addressed in the rebuttal.
Reviewer e29o would keep the same, as 10 was given already.
Reviewer KwhW would likely keep their score. The reviewer leaned toward acceptance to begin with, and the questions raised seem minor and would not be main decision changing factors.

---

### Decision · Program_Chairs · 2026-01-26

Accept (Poster)